# Effects of the Selective Decontamination of the Digestive Tract (SDD) on Pulmonary Secondary Infections in Patients with COVID-19 Acute Respiratory Distress Syndrome: A Retrospective Single Centre Experience

Giorgio Berlot [1] , Edoardo Moro [1,*] , Stefano Zio [2], Silvia Zanchi [1], Anna Randino [1] and Ariella Tomasini [1]

[1] Department of Anesthesia and Intensive Care, Cattinara University Hospital, Azienda Sanitaria Universitaria Giuliano Isontina—ASUGI, 34149 Trieste, Italy

[2] Faculty of Medicine, Department of Medicine, Surgery and Health Scineces, University of Trieste, 34149 Trieste, Italy

*   Correspondence: edoardo.moro@live.com; Tel.: +39-040-3994229

**Abstract:** Definitive data on the incidence rate of ventilator-associated pneumonia (VAP) in COVID-19 are still lacking, ranging from 29 to 58%. To date, most of the existing literature refers to patients who are not subjected to VAP prevention with selective decontamination of the digestive tract (SDD). We retrospectively collected data on all COVID-19 patients admitted to our ICU during the second phase of the pandemic with the aim of assessing the occurrence of VAP and the related mortality at 30 days and comparing our findings with the available literature. Of 213 patients, only 74 were eligible for the analysis. An incidence of 6.90 VAP per 1000 days of mechanical ventilation was detected. Apart from a smoking habit (0% vs. 10%, $p < 0.005$) and diabetes (14% vs. 54%, $p = 0.026$), patients who developed VAP did not differ significantly from those who did not regarding comorbidities, steroid use, and the severity of COVID-19. VAP were predominantly caused by mono-microbial Gram-negative or fungal infections. Mortality was significantly higher in those who developed VAP (86 vs. 33%, $p = 0.002$). Our evidence aligned with the available literature in assuming a possible role of SDD in reducing the incidence of VAP in COVID-19 patients, with a possible impact on related mortality and costs.

**Keywords:** selective decontamination of the digestive tract; ventilator-associated pneumonia; pulmonary infections; COVID-19; acute respiratory distress syndrome; mechanical ventilation

## 1. Introduction

Independently from the underlying disease, mechanically ventilated patients are prone to develop ventilator-associated pneumonia (VAP) caused by bacteria, fungi, and viruses. A number of conditions account for this finding, including tracheal intubation, prolonged sedation and/or muscle relaxation, lung tissue damage, the presence of secretions, alterations of the microbiome, hygiene conditions, the nurse-to-patient ratio, and others [1,2]. According to a largely accepted model, the more frequent microorganism(s) that are responsible for VAP are bacteria of the oropharyngeal flora, predominantly Gram-negative, which migrate downward into the respiratory tract and invade the lower airways [1]. Fungi rarely cause VAP, with *Candida* spp. being the most common isolated yeast (though the available data do not support a direct role of *Candida* spp. as a sole VAP-causative pathogen), but also with Aspergillus spp (in particular in patients affected by influenza) [1]. Viruses such as influenza, respiratory syncytial virus, and others may also be responsible for VAP. In particular, Herpes simplex virus (HSV) and Cytomegalovirus can be a source of VAP by reactivation in both immunocompromised and non-immunocompromised mechanically ventilated patients, possibly due to the immunoparalysis that characterizes intensive care unit (ICU) patients [1,2].

Although the effects of VAP on the outcome depend on many other factors, such as age and related frailties, the disease for which the patient is in the intensive care unit (ICU), or the immune status, it is clear that its occurrence prolongs the mechanical ventilation (MV), the length-of-stay in the ICU ($LOS^{ICU}$) and in the hospital ($LOS^{Hosp}$), and increases the overall costs of care. Both nonpharmacologic and pharmacologic strategies have been implemented to reduce the incidence of VAP, the first including the elevation of the head and the aspiration of the subglottic secretions, whereas the others take advantage of mouthwash with antiseptic substances and of the selective decontamination of the digestive tract (SDD) [1]. SDD consists of the administration of a short course of a systemic antibiotic during the initial days of MV (to prevent early infections with potentially pathogenic microorganisms that might be present in the patients' commensal flora), associated with topical and enteral non-absorbable antibiotics and antifungal agents throughout its entire duration (aiming at the eradication of the pathogens that overgrow during a critical illness from pharynx, upper airways, and gut, and the preservation of the endogenous anaerobic flora) [3–5]. Indeed, the antibiotics that are present in the SDD are mainly directed against aerobic Gram-negatives, as these account for the vast majority of VAPs and do not affect the normal anaerobic enteric microbiome, to prevent colonization with opportunistic pathogens [1,3].

It appears that patients with viral pneumonia are particularly susceptible to secondary lung infections. Indeed, in some studies involving critically ill patients, the rate of bacterial co-infection ranged from 11 to 35% [6,7], and an even higher incidence has been observed in patients with COVID-19 pneumonia and/or acute respiratory distress syndrome (ARDS) [8–10]. Although the underlying mechanisms are not fully clear yet, it has been hypothesized that several factors account for this finding, including a more severe damage of the alveolar epithelial lining, the impairment of the local antibacterial protective factors (such as interferon-2), and the effects of immunomodulating agents used to abate the COVID-19-associated cytokine release, including steroids and tocilizumab [8–10].

Most of the existing literature on VAP in COVID-19 refers to critical patients that are not subjected to SDD. Since in our ICU, SDD has been adopted since the 1980s and is considered a standard of care for VAP prophylaxis, the aim of our study was to assess the occurrence of VAP, evaluate the outcome of the COVID-19 patients in our ward, and compare these findings with those presented in the available literature.

## 2. Materials and Methods

A retrospective analysis was performed considering the cohort of patients admitted to our dedicated COVID-ICU with a diagnosis of COVID-19-related pneumonia or ARDS requiring MV between 1 October 2020 and 31 March 2021. The SARS-CoV-2 infection was confirmed by a reverse transcriptase polymerase chain reaction (PCR) performed on a broncho-alveolar lavage (BAL) obtained at admission. Eligible criteria included MV for >48 h while in ICU. Patients with pre-existing immunodepression (i.e., chronic corticosteroid or autoimmune disease) and transfer from another ICU not using SDD after >48 h of MV were excluded from the study.

To note, the overall treatment followed the current guidelines on ARDS [11] and most of the relevant evidence on COVID-19 [12–15].

In light of the retrospective nature of the study and the use of SDD as a standard of care in our ICU, with respect to patients' privacy and the exclusive use of data that were normally collected for routine clinical practice, an ethical committee's approval and a patient's informed consent were not required. All data were collected and described anonymously. Informed consent for general hospital care is routinely signed by all patients that are admitted to the hospital, whether conscious or unconscious, who have activated the emergency system.

### 2.1. Selective Digestive Decontamination

The SDD protocol consisted of (a) a preparation containing tobramycin, polymyxin E, and amphotericin B administered four times a day along the entire duration of the MV as a paste in the oropharynx and as a solution in the enteric tract via a gastric tube; (b) a third-generation cephalosporin (cefotaxime in previously healthy patients, ceftazidime in patients with preexisting chronic diseases, or commonly accepted risk factors for multidrug-resistant pathogens [16–18]) given intravenously for the initial 5 days of MV. The paste and the solution were prepared in the hospital pharmacy and the overall cost of the treatment (paste and solution and parenteral antibiotics) was approximately EUR 35 per patient per day (~USD 36). Throat and rectal swabs were obtained at admission before the first administration of SDD and twice a week thereafter.

### 2.2. Ventilator-Associated Pneumonia Definition

Due to the difficulty in diagnosing VAP relying strictly on commonly accepted criteria [18,19] (i.e., a combination of radiological, clinical, and microbiological criteria), in the context of an ARDS (e.g., new and persistent pulmonary infiltrate on chest imaging, leukopenia/leukocytosis, fever, or worsening of gas exchange, possibly due to both VAP or worsening of COVID-19 ARDS itself) [20–22], and to avoid an overestimation, we diagnosed VAP only if a new respiratory pathogen was detected in a significant quantitative growth in the fluid retrieved with a BAL fluid. Clearly, BAL execution was triggered by clinical suspicion of superinfection (e.g., evidence of worsening of gas exchanges or respiratory mechanics despite optimization of ventilation, and/or sudden increase in respiratory secretion, and/or increase in the white blood cell (WBC) count, and/or increase in serum C-reactive protein, and/or increase in serum procalcitonin, and/or sustained fever, and/or sudden increase in vasopressor requirements). BAL was defined as positive in case of the following: $\geq 104$ colony-forming units/mL for bacteria and positive PCR for viruses or fungi [19].

### 2.3. Data Collection

A number of different types of information were retrieved from the medical records, including (1) at ICU admission: age, sex, comorbidities, Charlson comorbidity index, SAPS II and SOFA scores, the days elapsing from the onset of symptoms and the tracheal intubation, the duration of steroids given as a part of the COVID-19 treatment, and the antibiotics received before VAP onset (not including those used for the SDD); (2) the time intervals between the diagnosis of COVID-19 infection, the LOS$^{Hosp}$ and LOS$^{ICU}$, and the diagnosis of VAP; (3) the microbiological findings; and (4) the duration of the MV, the overall LOS$^{ICU}$, and the outcome at the 30th day. The primary outcome was VAP incidence, and the secondary was 30-day mortality.

### 2.4. Statistical Analysis

Data were expressed as mean and interquartile range (IQR) or number and proportion (%) for continuous and categorical variables, respectively. The Mann–Whitney U test and Chi-square or Fisher exact tests were used for the baseline characteristics analysis. Comparison by univariate Cox proportional hazards model was used to evaluate the development of VAP, which was censored for extubation and death. The incidence rate of VAP was expressed as the number of events per 1000 ventilator days and was compared using the Mid-P exact test. Independent predictors of development of VAP were evaluated using univariate logistic regression analyses and described as odds ratio, confidence interval (CI) of 95%, and a p-value. Finally, risk factors for VAP were compared using a Cox proportional hazards model. A *p*-value $\leq 0.05$ was considered statistically significant. All the analyses were performed with IBM SPSS Statistics for Windows, Version 26.0 (IBM Corp, Armonk, NY, USA).

### 3. Results

During the study period, 213 not-vaccinated adult patients with COVID-19-related ARDS were admitted to our dedicated ICU. Ninety-five of them were excluded from the analysis due to unavailability of all the data, and another forty-four fulfilled the exclusion criteria. The remaining 74 patients were included in the study (Table 1).

**Table 1.** Demographic data and other characteristics at the ICU admission. BMI: Body Mass Index; SAPS II, Simplified Acute Physiology Score II; SOFA, Sequential Organ Failure Assessment; CCI: Charlson's Comorbidity Index; WBC: white blood cell count; IL-6: Interleukin 6; CRP: C-reactive protein; VAP: patients who developed a VAP; non-VAP: patients who did not develop a VAP.

| | All (*n* = 74) | VAP (*n* = 7) | Non-VAP (*n* = 67) | *p* |
|---|---|---|---|---|
| Age (IQR) | 73 (66–77) | 71 (69–73) | 73 (65–77) | 0.197 |
| Sex (% female) | 18 (24) | 2 (29) | 16 (24) | 0.361 |
| BMI (IQR) | 29 (26–32) | 32 (27–33) | 29 (26–32) | 0.292 |
| Days from infection (IQR) | 8 (5–13) | 9 (6–13) | 8 (5–12) | 0.459 |
| Pre-ICU steroid use (%) | 37 (50) | 4 (57) | 33 (49) | 0.535 |
| Pre-ICU steroid duration in days (IQR) | 6 (3–10) | 4 (3–5) | 7 (3–10) | 0.116 |
| Pre-ICU antibiotic use (%) | 39 (53) | 5 (71) | 34 (51) | 0.559 |
| Co-infection (%) | 22 (30) | 0 (0) | 22 (33) | 0.053 |
| SAPS II (IQR) | 42 (37–45) | 40 (36–44) | 42 (37–45) | 0.951 |
| SOFA score (IQR) | 6.50 (4–9) | 5.00 (4–5) | 7.00 (4–9) | 0.064 |
| CCI (IQR) | 4 (3–3) | 4 (3–3) | 4 (3–3) | 0.729 |
| WBC (IQR) | 12 (8–17) | 12 (8–15) | 12 (9–17) | 0.663 |
| IL-6 (IQR) | 59 (28–122) | 48 (13–71) | 59 (28–126) | 0.769 |
| CRP (IQR) | 131 (88–194) | 138 (71–184) | 130 (88–194) | 0.508 |

At ICU admission, 29.73% of patients presented a concomitant infection (45.5% airways, 36.4% bloodstream, 4.54% urinary tract, and 13.63% other source), which resulted in a bacterial infection in 90.9% of cases. The median interval between COVID-19 symptoms onset and ICU admission was 8 (5.00–12.75) days. A total of 50% of the subjects analyzed were taking corticosteroid upon admission to the ICU as a part of therapy for COVID-19 for a median of 6 (3–10) days. Apart from steroids, only three patients were subjected to an immunomodulatory therapy, respectively, one to IL-6 antagonism with tocilizumab, and two to cytokine removal with a hemadsorption technique (all three did not develop a VAP episode). Overall, seven patients developed a VAP, corresponding to 6.90 (3.60–14.90) events per 1000 ventilator days, after a median of 8 (5.50–11.50) days of MV. The time intervals preceding the diagnosis of VAP were as follows: 18 (14–25) days from the diagnosis of SARS-CoV-2 infection, 12 (9–15) days of LOS$^{Hosp}$, 8 (6–12) days of LOS$^{ICU}$, and 8 (6–12) days of MV.

Regarding the duration of corticosteroid therapy, no differences were found between VAP and non-VAP patients (12 (10.50–13.75) vs. 12 (10.00–14.00) days, *p* = 0.837). The two groups did not differ in terms of frailty and severity scores or inflammatory status (Table 1). Apart from the smoking habit (0 (0%) vs. 7 (10%), *p* < 0.005) and diabetes (1 (14%) vs. 36 (54%), *p* = 0.026), the rate of different comorbidities did not differ significantly between VAP and non-VAP patients (Table 2).

The two groups were comparable in terms of LOS$^{ICU}$ and LOS$^{Hosp}$ and the duration of the MV (Table 3).

VAP were predominantly caused by a single bacterial super-infection (namely *E. cloacae* 25%, *K. aerogenes* 12.5%, and *S. aureus* 12.5%). One patient developed a mixed infection by *K. pneumonia* and *A. fumigatus*. Notably, there were three cases that were induced by a single non-bacterial pathogen (*A. fumigatus*, *C. albicans*, and herpes simplex virus type 1). No multi-resistant rods were discovered. The overall mortality rate at 30 days was 38%, and it was significantly higher in VAP than in non-VAP patients (86 vs. 33%, *p* = 0.002). Despite an apparent trend, age was not significantly lower in patients who deceased with VAP (70, IQR 68–70 years) as compared with those who deceased without VAP (73, IQR 68–75 years).

**Table 2.** Patients' comorbidities at admission in intensive care unit. VAP: patients who developed a VAP; non-VAP: patients who did not develop a VAP.

| | All (*n* = 74) | VAP (*n* = 7) | Non-VAP (*n* = 67) | *p* |
|---|---|---|---|---|
| Neurological (%) | 19 (26) | 2 (29) | 17 (25) | 0.963 |
| Cardiovascular (%) | 54 (73) | 6 (86) | 48 (72) | 0.331 |
| Respiratory (%) | 15 (20) | 2 (29) | 13 (19) | 0.726 |
| Renal (%) | 9 (12) | 0 (0) | 9 (13) | 0.268 |
| Gastrointestinal (%) | 11 (15) | 1 (14) | 10 (15) | 0.843 |
| Hepatic (%) | 7 (9) | 0 (0) | 7 (10) | 0.336 |
| Coagulative (%) | 3 (4) | 0 (0) | 3 (4) | 0.541 |
| Diabetes Mellitus (%) | 37 (50) | 1 (14) | 36 (54) | 0.026 |
| Neoplasm (%) | 1 (1) | 0 (0) | 1 (1) | 0.728 |
| Smoke (%) | 7 (9) | 0 (0) | 7 (10) | <0.005 |

**Table 3.** Steroid use, length of stay, and mechanical ventilation during intensive care unit stay and outcome after 30 days. ICU: intensive care unit; VAP: patients who developed a VAP; non-VAP: patients who did not develop a VAP.

| | All (*n* = 74) | VAP (*n* = 7) | Non-VAP (*n* = 67) | *p* |
|---|---|---|---|---|
| Steroids use in ICU (%) | 69 (93) | 7 (100) | 62 (93) | 0.423 |
| Days of steroids in ICU (IQR) | 9 (5–12) | 10 (8–11) | 9 (5–12) | 0.499 |
| Total days of steroids use (IQR) | 12 (10–14) | 12 (10–14) | 12 (10–14) | 0.837 |
| ICU length of stay (IQR) | 13 (7–20) | 20 (14–20) | 12 (6–19) | 0.153 |
| Duration of mechanical ventilation (IQR) | 12 (6–20) | 20 (14–20) | 12 (6–17) | 0.111 |
| ICU mortality at 30 days (%) | 28 (38) | 6 (86) | 22 (33) | 0.002 |

## 4. Discussion

The occurrence of VAP is common in mechanically ventilated patients, with an incidence rate ranging from 5 to 40% [1]. Different factors account for this wide epidemiological variability, including the underlying conditions, the time window of the diagnosis, and the diagnostic criteria used [1]. In immunologically competent subjects, these include clinical suspicion, the appearance of new and persistent infiltrates on the chest radiographs, and the microbiological findings on samples drawn from the lower respiratory tract [1,18,19]. However, in patients with COVID-19-associated ARDS, the first two criteria appear unsuitable due to both the clinical conditions and the widespread involvement of the lungs, making a new opacity hard to identify from the background on a chest radiograph or a CT scan [20–22]. Indeed, the microbiological criteria were used in all the most relevant studies, albeit applying different threshold values to establish the diagnosis of VAP [9,10,22–28].

In COVID-19 patients that developed pneumonia and/or ARDS, it appears that secondary pulmonary infections (a) occur more frequently as compared with other viral pneumonia [8,25,29–31]; (b) are caused primarily by Gram-negative bacteria and fungi [8,29–34]; and (c) are associated with a longer MV [8,25,32]. Nevertheless, their effect on the outcome is not clear, being worse in some studies but not in others [22,31,32]. Similarly to what has been reported in other clinical scenarios, the SDD has not been widely used in the COVID-19 population. Notwithstanding several decades of use and different clinical trials, reviews, and meta-analyses [35–37] that have demonstrated a reduced incidence of VAP in patients receiving the SDD, its use is still limited, which is mainly due to the fear of the appearance of strains that are resistant to its components and to economic reasons [3,38–40]. Both issues are not supported, as different investigations have demonstrated that (a) there was not an increased trend in the antibiotic resistance in blood isolates; (b) in respiratory isolates, the resistance to the third-generation cephalosporin (cefotaxime and ceftriaxone) increased in patients not receiving but decreased in those treated with SDD; and, finally, (c) the preparation is not expensive and its use has been associated with reductions in antibiotic costs [3,37,41–43]. A recently published case–control study [44] analyzed the

incidence of multidrug-resistant (MDR) bacteria and VAP over a 5-year period in 11 German ICUs using (case group) or not using (control group) a selective decontamination of the oropharynx (SOD—a conceptually similar approach to VAP prevention). Among the 5034 patients involved in the final analysis, the rates and incidence densities of all ICU-acquired MDR bacteria did not differ between the two groups, except for a significantly increased incidence of vancomycin-resistant Enterococcus faecalis (VRE), and a significantly decreased incidence of extended-spectrum beta-lactamase-producing Klebsiella pneumoniae in the SOD group. Furthermore, an increased antibiotic selection pressure under SOD was not detected. Interestingly, during the study period, the prevalence of VRE increased in German hospitals. A lower incidence density of VAP was recorded in patients receiving SOD (10.2 vs. 14.1/1000 days of MV, $p < 0.01$), but not for bacteremia or urinary tract infections. Finally, the incidence rate of death revealed fewer deaths in the ICU in patients treated with SOD (23 vs. 30%, $p < 0.01$), also after propensity score matching (28 vs. 30%, $p < 0.01$). Similarly, in a randomized trial conducted in 13 European ICUs [45], SOD and SDD were not associated with a reduction in ICU-acquired bloodstream infection (BSI) caused by MDR Gram-negative bacteria, nor mortality, or increased prevalence of carriage with antibiotic-resistant bacteria compared to standard care (daily chlorhexidine body wash and hand hygiene program). Albeit not designed as a head-to-head comparison, SDD was associated with a lower risk of ICU-acquired MDR Gram-negative bacteria BSI compared with chlorhexidine mouthwash on a post hoc analysis. Furthermore, a large crossover, cluster randomized clinical trial [40] demonstrated a significant reduction in positive blood cultures and cultures of antibiotic-resistant organisms and no significant increase in new *Clostridium difficile* infections in patients who received SDD, which was associated with a decrease in overall antibiotic use. Our results seemed to confirm this trend, as we did not report MDR bacteria in respiratory tract isolates.

As far as the issue of SDD in COVID-19 is concerned, it must be remarked that, to the best of our knowledge, there are not double-blind clinical trials assessing its effects. With this relevant limitation, it seems that the effect of the SDD on the incidence of VAP also applies to patients with COVID-19 ARDS. A recent Dutch single-center survey [23] comparing the incidence of VAP in COVID-19 patients receiving SDD as a standard of care demonstrated a lower rate of secondary bacterial pulmonary infections (10%) as compared with that observed by other investigators not using it [9,10,22,24–28], ranging from 29 to 58%. Louque-paz et al. [46] reported the results of a retrospective observational study comparing two French ICUs, one in which SDD was routinely administered, and one that did not use any sort of antibiotic prophylaxis of VAP. VAP incidence was lower in the SDD group than in the group without SDD (9.4 vs. 23.5 per 1000 ventilator days, $p < 0.001$), as also confirmed by the Log-rank test ($p < 0.001$) and the multivariate analysis (adjusted HR 0.36; 95% CI [0.20–0.63]). In addition, mortality rates at day 28 were lower in the SDD group (adjusted HR 0.33, 95% CI [0.12–0.87]; $p = 0.03$) [46]. In a pre–post study conducted in Italy [47], the use of SDD in a structured protocol for VAP prevention in COVID-19 ARDS seemed to reduce the occurrence of VAP (26.8 vs. 20.4%, mainly late VAP) at the multivariate analysis (HR 0.536, CI 0.338–0.851; $p = 0.008$). Interestingly, the onset time of VAP (median time of 8 days after initiation of MV), the occurrence of MDR microorganisms, the length of invasive MV, and hospital mortality were similar in the patients who received and who did not receive SDD. Moreover, in a retrospective analysis of 15 ICUs, Massart et al. [48] reported a lower incidence (14.3 vs. 28.3 per 1000 ventilatory days, $p < 0.001$) and a lower risk of VAP (incidence rate ratio 0.52 (0.38–0.83), $p = 0.005$) in patients subjected to multiple-site decontamination (MSD, SDD plus chlorhexidine body washing, and a nasal mupirocin course), but not of BSI; no difference in bacterial antibiotic-resistance pattern was reported between the MSD group and the control group [47]. A brief summary of the cited studies is shown in Table 4.

**Table 4.** Result of other studies on ventilator-induced pneumonia in COVID-19 pneumonia or ARDS. § Referred to patients subjected to selective decontamination of the digestive tract only (see the reference for details). * Referred to patients subjected to multiple-site decontamination only (see the reference for details). CFU: Colony-forming units; BAL: broncho-alveolar lavage; TBA: tracheo-bronchial aspirate; SDD: selective decontamination of the digestive tract; VAP: ventilator-induced pneumonia; n.a.: not available; n.r.: not reported; s.c.: single center; m.c.: multi center.

| Authors | Study Design | Cutoff Value (CFU/µL) | | SDD | Number of Patients | VAP Incidence (%) | VAP per 1000 Ventilator Days | VAP Mortality (%) |
|---|---|---|---|---|---|---|---|---|
| | | TBA | BAL | | | | | |
| Blonz, G. et al. [24] | m.c. | $\geq 10^3$ | $\geq 10^4$ | no | 188 | 48.9 | 39 | 31 |
| Rouzè, A. et al. [25] | m.c. | $\geq 10^5$ | $\geq 10^4$ | no | 568 | 50.5 | n.r. | 29 |
| Maes, M. et al. [26] | s.c. | $\geq 10^5$ | $\geq 10^4$ | no | 94 | 39.0 | 28 | 38 |
| Pickens, C.O. et al. [9] | s.c. | n.a. | $\geq 10^4$ | no | 179 | 44.4 | 45.2 | 11 |
| Vacheron, C.H. et al. [10] | m.c. | n.r. | n.r. | no | 1879 | 29.0 | 25.5 | 31 |
| Giacobbe, D.R. et al. [22] | m.c. | n.r. | n.r. | no | 586 | 29.0 | 18 | 46 |
| Nseir, S. et al. [27] | m.c. | $\geq 10^5$ | $\geq 10^4$ | no | 568 | 36.0 | n.r. | 26 |
| COVID-ICU Group [28] | m.c. | n.r. | n.r. | no | 2101 | 58.0 | n.r. | n.r. |
| Van der Meer, S.B. et al. [23] | m.c. | n.r. | n.r. | yes | 212 | 10.0 | n.r. | 41 |
| Biagioni et al. [47] | s.c. | n.r. | n.r. | yes | 86 § | 27 § | n.r. | 54 § |
| Massart et al. [48] | m.c. | n.r. | n.r. | yes | 89 * | 29 * | 14 * | 17 * |
| Luque-Paz et al. [46] | m.c. | n.r. | n.r. | yes | 77 § | 21 § | 9 § | 7 § |

Our study basically replicated and grossly reinforced these findings, as either the incidence and time interval from the diagnosis of COVID-19 and the occurrence of VAP largely overlap (late VAP, i.e., >96 h after the beginning of MV). As previously described, our cohort of patients with VAP were ventilated for a longer time and had a longer LOS$^{ICU}$ and LOS$^{Hosp}$ than non-VAP patients, generating a further potential increase in the stress on the health system and in the overall costs of care. Not surprisingly, their mortality rate was higher not only than that in non-VAP patients, but also than that reported in other studies involving similar patients. The baseline conditions were similar between VAP and non-VAP patients, apart from a more frequent smoking habit and diabetes in non-VAP patients, despite them being known as possible risk factors for VAP [49].

Finally, the protective effect of SDD could not be limited to the lungs, as a single-center survey recently demonstrated that critically ill COVID-19 patients given SDD did not develop candidemia [50], as compared with an incidence rate of 0.8–14% observed in untreated patients [51].

As previously declared, the main limitation of this single-center retrospective study could be the lack of a control group not receiving SDD. Therefore, a causal relationship between the use of SDD and the incidence of VAP cannot be clearly established. The absence of a subgroup not subjected to SDD, conditioning the design of the study and somewhat underpowering our results, relies on the consolidated practice of using SDD in all patients subjected to MV for more than 48 h as part of a standardized internal protocol of care in our ICU since the 1980s (a practice reinforced over time thanks to the growing amount of literature existing). Not administering this therapy to patients would have represented a lowering of the essential therapeutic standards. Furthermore, retrospectively identifying VAPs solely based on microbiological criteria led to a risk of bias and consequently underdiagnosis, albeit the respiratory tract samples were always driven by clinical suspicion of superinfection. Lastly, the little sample size could represent another limitation of our analysis, in which the possible effect of a bias was impossible to remove due to the retrospective nature of the analysis, although our results seemed to reproduce those of other works on this topic.

## 5. Conclusions

Mechanically ventilated patients are prone to developing VAP caused by several bacteria, viruses, and fungi. Not surprisingly, this also applies to patients with COVID-19 pneumonia, which is likely due to the extensive damage to the alveolar-epithelial lining and to the strong local inflammatory reaction associated with the multifactorial reductions in the immune capabilities. Although no clear causal relationship could be established,

principally due to the lack of a control group, our evidence seems to align with other evidence derived from the available literature in assuming a possibly reduced incidence of VAP with SDD in patients affected by COVID-19-induced pneumonia or ARDS and maybe, more generally, in critically ill patients. Clearly, this evidence and the possible contribution to reducing VAP incidence and related mortality require further rigorous studies to clarify the associations.

**Author Contributions:** Conceptualization, G.B. and E.M.; methodology, G.B. and E.M.; formal analysis, S.Z. (Stefano Zio); investigation, S.Z. (Stefano Zio) and S.Z. (Silvia Zanchi); data curation, S.Z. (Stefano Zio), E.M. and S.Z. (Silvia Zanchi); writing—original draft preparation, G.B. and E.M.; writing—review and editing, G.B., E.M., A.R. and A.T.; visualization, E.M.; supervision, G.B., E.M., A.R. and A.T.; project administration, G.B. All authors have read and agreed to the published version of the manuscript.

**Funding:** This research received no external funding.

**Institutional Review Board Statement:** An ethical committee's approval was not required due to the nature of the study, the use of the SDD as a standard of care in daily practice, and the exclusive use of data that were normally collected for routine practice.

**Informed Consent Statement:** In light of the retrospective nature of the study, the use of SDD as a standard of care in our ICU, and the exclusive use of data that were normally collected for routine practice, with respect to patients' privacy a patient's informed consent was not required. Furthermore, informed consent for general hospital care is routinely signed by all patients that are admitted to the hospital, whether conscious or unconscious, who have activated the emergency system.

**Data Availability Statement:** The datasets used and analyzed during the current study are available from the corresponding author on a reasonable request. The data are not publicly available due to privacy.

**Conflicts of Interest:** The authors declare no conflict of interest.

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
