# Peer review of "Effects of the Selective Decontamination of the Digestive Tract (SDD) on Pulmonary Secondary Infections in Patients with COVID-19 Acute Respiratory Distress Syndrome: A Retrospective Single Centre Experience"

_gastrointestdisord, doi:10.3390/gidisord5020019_

Round 1

Reviewer 1 Report

Dear Editor of Gastrointestinal Disorders,

I deeply appreciate the opportunity to review the manuscript titled “Effects of the Selective Decontamination of the Digestive Tract (SDD) on Pulmonary Secondary Infections in Patients with COVID-19 ARDS: a Single Centre Experience”. The authors presented a retrospective cohort study that aimed to “assess the occurrence of VAP and to evaluate the outcome of the COVID-19 patients” in their clinical setting, comparing their findings with those presented in the available literature.

The manuscript generally adheres to the journal template, although the formatting needs to be revised. It is also concise, with only a few grammar and syntax errors that can be addressed by the authors. However, I have some specific comments and/or concerns regarding the manuscript, which are outlined below:

[Title, abstract, keywords]

- Please revise the use of non-specified acronyms;

- As per international recommendations, the study design should be indicated in the title;

[Background]

The background section requires extensive revisions to properly introduce the need for this study. While the authors suggest that most existing literature on ventilator-associated pneumonia (VAP) in COVID-19 pertains to critical patients not subjected to selective digestive decontamination (SDD), this claim is vague and not entirely accurate. A quick search of a single database yielded several studies on the same topic, including those listed here:

https://pubmed.ncbi.nlm.nih.gov/34844031/

https://www.mdpi.com/2077-0383/12/4/1432

https://ccforum.biomedcentral.com/articles/10.1186/s13054-021-03869-y

https://hal.science/hal-03468436/document

Therefore, I strongly recommend that the authors provide a more comprehensive introduction to the topic, outlining the current gaps in the literature and explaining how their retrospective cohort study addresses them.

[Methods]

- This section would greatly benefit from the use of subheadings such as "Study Design and Setting," "Sample and Recruitment," "Outcomes and Instruments," and "Data Analysis" to enhance readability.

- Please clearly identify how recruitment and screening was conducted, given the retrospective nature of the study;

- Clearly define all outcomes, exposures, predictors, and potential confounders. Some diagnostic criteria is not supported by a reference;

- The authors claim that “an ethical committee approval and a patient’s informed consent were not required”. While ethical committee approval might have been waived, the authors should still identify what procedures were in place to guarantee data privacy and anonymity.

[Results]

- Please consider revising the sentence starting on line 151 of page 4.

- It is recommended that Table 2 be converted to text format since the type and volume of data does not require a table.

- The sequence of text and tables should be revised. For example, on page 5, line 167, the authors refer back to Table 1, despite already introducing Table 2.

- On page 5, lines 166-167, the authors state that "The two groups did not differ in terms of frailty and severity scores, and inflammatory status (Table 1)." Since frailty is a well-studied concept, it is unclear what the authors mean without a clear definition and explanation of the instruments used in the Methods section.

- Figure 1 may not add value to the manuscript and could be removed.

- Please italicize the names of all bacteria.

- Table 5 may not be relevant in the Results section and should be integrated into the discussion.

[Discussion and conclusions]

-        Given that smoking and diabetes have been linked to a higher risk and recurrence rate of upper respiratory tract infections, it would be beneficial for the authors to provide a more detailed discussion on the differences between the study samples.

-        Please revise the limitations section. A retrospective observational study should not necessarily be considered limited due to the absence of a control group, as this is a characteristic of this type of study design.

-        The authors should address any potential interpretation bias arising from the sample sizes and describe the measures taken to address potential sources of bias.

Reviewer 2 Report

Thanks for giving me the opportunity to review this article. It is a monocentric retrospective study, focusing on the occurrence of VAP and associated factors of VAP in an ICU that use Selective decontamination of the digestive-tract (SDD) as a VAP prevention protocol. The aim of this study was to compare their epidemiology to the epidemiology reported in the literature, underlying that in most previous studies, SDD is not use.

To my mind, this study as numerous limits, most of all from a methodological point and deserve major modifications to be considered.

My main concern relies on the design. It is not possible to say that SDD prevent from VAP and risk of death if you do not have direct comparison (i.e. patients without SDD, in the same cohort). For that purpose, you are not allowed to have this aim, nor your conclusion. There is to many other confounding factors such as severity of the patients, comorbidities, burden for caregivers, COVID-19 treatments that should be considered to explain your prevalence of VAP and your mortality.

Then, if you consider only the risk of VAP and occurrence of VAP, there is now a lot of published studies on that subject.

The methodology to explore the risk of VAP should be improved (uni and then multivariate analyses with the Cox model, and even more, you could consider the competing risks of death and or living alive from ICU).

Because your methodology can't be accepted, aims and conclusions of the study should be changed and the paper should only focus on incidence and risk of VAP among COVID 19 patients.

Reviewer 3 Report

This article was exceptionally well written. It claims that decontamination of digestive tact could reduce the incidence of VAP and related mortality in mechanically ventilated patients COVID 19 patients.

There are multiple studies in the literature that have discussed the use of SDD associated with reduction of pneumonia but SDD is still sporadically used, so this article could help clinicians to make decisions about the usage of SDD especially in COVID 19 patients. Author still has to explain the scientific gap. This article claims that it has shown reduced mortality, however methodology and conclusions needs to be addressed.  Authors can use sub-headings like treatment protocol ( duration), data collection and data analysis. Need more clarification regarding recruitment of patients into the study. Consider using univariate analysis and adjusted analysis with confounders.  As methodology have to be revised, conclusion needs to be addressed.

It would be intriguing if author can discuss why they used  antibacterial and antifungal but not antiviral in SDD protocol even though the percentage of HSV 1 responsible for VAP is 12.5 %

Table 5 can be switched to discussion paragraph

Figure 1 can be removed or author can explain why he hasn't used antivirals in spite of HSV 1 incidence of 12 %  

Round 2

Reviewer 1 Report

Dear Authors,

I would like to express my satisfaction with the restructuring of the manuscript in response to the provided feedback. The scientific rigor and transparency in reporting your findings have improved, particularly in light of your comments on the absence of a definitive causal relationship between VAP in COVID-19 patients and SDD. However, I have noted that these improvements are not apparent in the manuscript's abstract. Therefore, I kindly request that you revise the abstract accordingly.

Finally, I have noticed that the latest revision of the manuscript has a layout that does not meet the journal's requirements, such as tables and reference style. However, I am confident that these issues can be resolved during the post-production editing process.

Reviewer 2 Report

All the modification have been taken into account

A description of the excluded patients because of missing data should be reported.

Once again, no conclusion can be drawn from this work because of the absence of group control
